# Paternal low protein diet perturbs inter-generational metabolic homeostasis in a tissue-specific manner in mice

Hannah L. Morgan[1], Samuel Furse [2,3,4], Irundika H. K. Dias[5], Kiran Shabir[5], Marcos Castellanos[6], Iqbal Khan[6], Sean T. May [6], Nadine Holmes[7], Matthew Carlile [7], Fei Sang [7], Victoria Wright[7], Albert Koulman [2,3] & Adam J. Watkins [1✉]

The underlying mechanisms driving paternally-programmed metabolic disease in offspring remain poorly defined. We fed male C57BL/6 mice either a control normal protein diet (NPD; 18% protein) or an isocaloric low protein diet (LPD; 9% protein) for a minimum of 8 weeks. Using artificial insemination, in combination with vasectomised male mating, we generated offspring using either NPD or LPD sperm but in the presence of NPD or LPD seminal plasma. Offspring from either LPD sperm or seminal fluid display elevated body weight and tissue dyslipidaemia from just 3 weeks of age. These changes become more pronounced in adulthood, occurring in conjunction with altered hepatic metabolic and inflammatory pathway gene expression. Second generation offspring also display differential tissue lipid abundance, with profiles similar to those of first generation adults. These findings demonstrate that offspring metabolic homeostasis is perturbed in response to a suboptimal paternal diet with the effects still evident within a second generation.

[1] Lifespan and Population Health, School of Medicine, University of Nottingham, Nottingham NG7 2UH, UK. [2] Core Metabolomics and Lipidomics Laboratory, Wellcome-MRC Institute of Metabolic Science, University of Cambridge, Addenbrooke's Treatment Centre, Keith Day Road, Cambridge CB2 0QQ, UK. [3] Wellcome-MRC Institute of Metabolic Science and Medical Research Council Metabolic Diseases Unit, University of Cambridge, Cambridge CB2 0QQ, UK. [4] Biological Chemistry Group, Jodrell Laboratory, Royal Botanic Gardens Kew, Richmond TW9 3AE, UK. [5] Aston Medical School, Aston University, Aston Triangle, Birmingham B4 7ET, UK. [6] Nottingham Arabidopsis Stock Centre (NASC), University of Nottingham, Sutton Bonington Campus, Plant Science Building, School of Biosciences, Loughborough LE12 5RD, UK. [7] Deep Seq, School of Life Sciences, Queen's Medical Centre, University of Nottingham, Nottingham NG7 2UH, UK. ✉email: adam.watkins@nottingham.ac.uk

There is now growing acceptance that a mother's diet during, and even before pregnancy can increase the risk that her offspring will develop a range of metabolic disorders[1]. Both maternal under- and over-nutrition during pregnancy can increase the risk of her adult offspring becoming overweight, glucose/insulin intolerant and developing dyslipidaemia[2]. Furthermore, impaired offspring metabolic health can be programmed over multiple generations, even when they themselves are not exposed to the same poor-quality diet[3–6].

Mirroring the programming effects of a poor maternal diet, paternal under- and over-nutrition around the time of conception have been shown to affect the cardiovascular and metabolic health of his offspring[7]. Underlying these programmed changes in offspring health are proposed sperm epigenetic and seminal plasma-specific mechanisms[8]. Here, perturbed patterns of sperm DNA methylation, histone modifications and/or non-coding RNA populations connect poor paternal diet at the time of conception with post-fertilisation development[9,10]. Furthermore, the composition of the seminal plasma and its influence on the pre-implantation uterine environment provides an additional paternal programming mechanism[11]. We have shown previously that both the sperm and the seminal plasma from male mice fed a sub-optimal low protein diet (LPD) can programme long-term offspring growth, metabolic[12,13] and cardiovascular[14] ill-health across two generations. Underlying the inter-generational programming were significant changes in the profile of gondal epigenetic regulators of DNA methylation, histone modifications and RNA methylation in first-generation offspring[14].

Central to the development of many metabolic diseases is imbalances in lipid metabolism, driven by perturbed cellular lipolysis and lipogenesis[15] and lipid traffic between tissues[12]. Excessive uptake and accumulation of triglycerides and non-esterified fatty acids in the liver are major drivers of non-alcoholic fatty liver disease (NAFLD) progression and liver damage[16]. Excessive amounts of fatty acids impair mitochondrial function, reducing their ability to perform beta-oxidation[17]. Furthermore, the accumulation of fatty acids has been shown to promote a pro-inflammatory status[18], fibrosis[19] and tissue damage[20] through the synthesis of pro-inflammatory prostaglandins, thromboxanes, and leukotrienes[21]. Separately, elevated levels of hepatic and plasma sphingolipids in mice have been associated with insulin resistance, tissue damage and dysfunction through the production of excessive ceramides[22]. In recent years, a new interaction between the microbiome, lipid metabolism and adult physiology has come to the forefront. The gut microbiome in both humans and mice appears able to both influence, and be influenced by, dietary composition and lipid abundance[23]. In mice, the gut microbiota of animals fed a diet high in saturated animal fat (lard) differed from those fed diets high in poly-unsaturated fish oil[24], associated with reduced insulin sensitivity. Interestingly, faecal transplants into germ-free mice identified the microbiome as an underlying mechanism driving the metabolic changes in these mice. Separately, acetate produced by the gut microbiota can be used by the liver in the synthesis of fatty acids as well as stimulating the liver's production of mono-unsaturated and poly-unsaturated fatty acids in mice[25]. Therefore, the gut microbiota could play a pivotal role in regulating lipid homeostasis and metabolic health.

In the current study, we examine the impact of poor paternal diet in mice on offspring tissue lipid metabolism to define when in postnatal life metabolic ill-health became manifest. Here, we observe age-specific changes in offspring tissue lipid abundance, underpinned by specific liver transcriptomic profiles. Interestingly, differential tissue lipid abundance was observed across generations, with second-generation offspring displaying profiles similar to those of first-generation adults.

## Results

**F1 neonatal offspring.** Previously, we have shown that offspring growth and cardiovascular health are impaired in response to a sub-optimal paternal low protein diet at the time of conception[13,14]. Furthermore, these changes were programmed through sperm and seminal plasma-specific mechanisms over two generations[13,14]. In the current study, we have used our mouse model, in which we combine artificial insemination with vasectomised male mating, to analyse the tissue lipid profile in four groups of offspring termed NN (NPD sperm and NPD seminal plasma), LL (LPD sperm and LPD seminal plasma), NL (NPD sperm and LPD seminal plasma) and LN (LPD sperm and NPD seminal plasma) at different ages and over two generations. Detailed descriptions of the stud male growth, reproductive fitness and litter sizes, as well as NN, LL, NL and LN offspring development, growth and cardiovascular phenotype, have been published[13,14].

To determine whether paternal diet programmed adult metabolic perturbations were established early in life, we conducted a detailed analysis of offspring lipid status at 3 weeks of age. As we observed no significant effect of offspring sex (Supplementary Fig. 1), data for neonatal males and females were combined. NL offspring displayed a significantly lighter body weight when compared to NN and LL offspring ($P < 0.05$; Fig. 1a). Analysis of offspring serum detected ~440 separate lipids in positive ionisation mode, comprising cholesteryl esters (CE), ceramides (CER), lyso-phosphatidylcholines (LPC), phosphatidylcholines (PC), phosphatidylethanolamine (PE), diglycerides (DG) and triglycerides (TG) (see Supplementary Data 2 for full profile). Initial principal component analysis (PCA) of differential lipids revealed no separation of treatment groups from each other in the serum (Fig. 1b). Analysis of the relative abundance of saturated, mono-saturated and poly-unsaturated lipids revealed no difference between groups (Fig. 1c). Analysis of individual serum lipids identified 9 with differential abundance (Fig. 1d) with all of them being different between the two 'mismatched' (NL and LN) groups and the non-mismatched groups (NN and LL). Similar to the changes seen in body weight, liver weight was significantly reduced in NL offspring when compared to NN and LL offspring ($P < 0.05$; Fig. 1e). As for the serum, all groups showed similar hepatic global lipid profiles (Fig. 1f, g). However, 12 lipids were identified with differential abundance, with 10 lipids being different between the LN or NL groups and the NN or LL groups. Three lipids SM-(42:00), TG-(46:03) and TG-(48:04), also showed a difference between NL and LN groups ($P < 0.05$; Fig. 1h). While there were no global influences on offspring sex on lipid abundance, 40 individual lipids were significantly different in their abundance between males and females (Fig. 1i, j) and were distributed across all the major lipid classes.

**Paternal diet modifies adult-offspring lipid profiles.** To determine whether the differences we observed in 3-week neonatal offspring were sustained into adulthood, we explored offspring phenotype and tissue lipid profiles at 16 weeks of age. LL, NL and LN offspring displayed a significantly heavier body weight at 16 weeks of age ($P < 0.05$; Fig. 2a), although the effect was smaller for females than males. The increased body weight, especially in LL, NL and LN males, was reflected in increased liver weight in all male groups compared to NN males ($P < 0.02$; Fig. 2b), while no difference was observed for females. No difference in mean gonadal fat pad weight was observed between groups (Fig. 2c). Initial analysis of hepatic free fatty acid levels revealed significantly elevated levels in NL and LN offspring when compared to NN and LN offspring ($P < 0.01$; Fig. 2d), with no effect of offspring sex observed. Subsequent analysis of hepatic triglyceride levels revealed no difference between groups (Fig. 2e). Similarly, no difference in circulating levels of

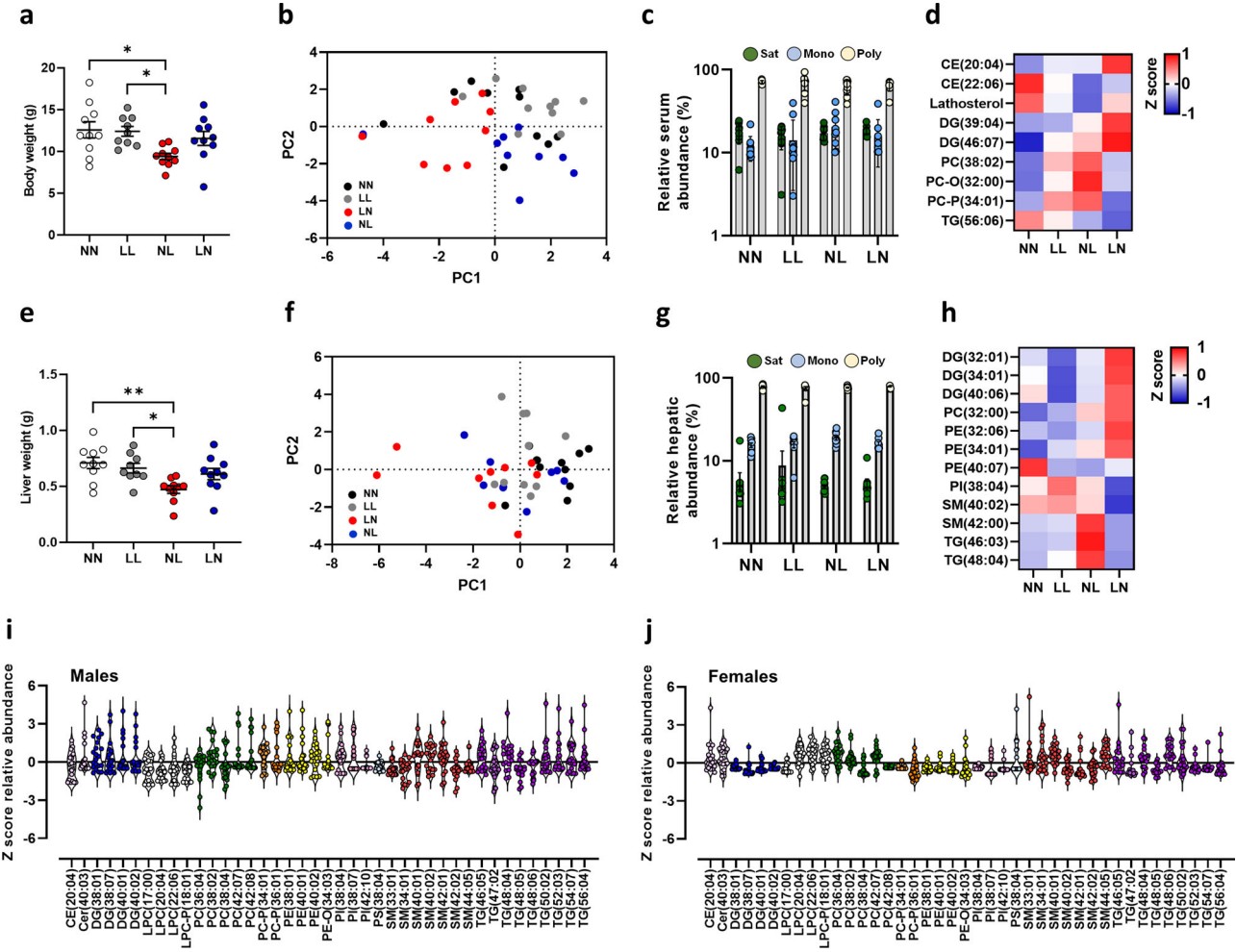

**Fig. 1 Impact of paternal diet on F1 neonatal offspring weights and tissue lipid profiles. a** Body weight of NN (NPD sperm and NPD seminal plasma), LL (LPD sperm and LPD seminal plasma), NL (NPD sperm and LPD seminal plasma) and LN (LPD sperm and NPD seminal plasma) offspring at 3 weeks of age. **b** Principal component analysis (PCA) of differential lipids in NN, LL, NL and LN offspring in serum. **c** serum abundance of saturated (Sat) mono-unsaturated (Mono) and poly-unsaturated (Poly) lipids. **d** Z scores of differential lipids between NN, LL, NL and LN offspring in serum. **e** liver weight of NN, LL, NL and LN offspring. **f** Principal component analysis (PCA) of differential lipids in NN, LL, NL and LN offspring liver tissue. **g** hepatic abundance of saturated (Sat) mono-unsaturated (Mono) and poly-unsaturated (Poly) lipids. **h** Z scores of differential lipids between NN, LL, NL and LN offspring in liver tissue. Cer ceramide, DG diglyceride (water-loss product from fragmentation in source), PC phosphatidylcholine, PC-P phosphatidylcholine plasminogen, PC-O phosphatidylcholine plasmalogen, PE phosphatidylethanolamine, PI phosphatidylinositol, SM sphingomyelin, TG triglyceride. Differential serum lipids in **i** male and **j** female offspring. $N = 9$–10 offspring (4–5 males and 5 females) per treatment group, sampled from all litters generated. Data are expressed as mean ± SEM (**a, b**). $*p < 0.05$, $**p < 0.01$. Statistical differences were determined using a random effects regression analysis (**a, b**) or one-way ANOVA with Bonferoni post hoc correction (**e, f**).

protein oxidation (measured as carbonyls) was observed between groups (Fig. 2f); however, NL and LN offspring displayed a lower level of circulating 8-isoprostane F2α when compared to NN offspring ($P < 0.02$; Fig. 2g).

Initial principal component analysis of lipid profiles in offspring liver, gonadal fat and serum (Supplementary Fig. 2) tissue show no significant separation between the four treatment groups. We also observed no significant separation based on sex (Supplementary Fig. 2). As such, data for male and female lipid profiles were combined. Similar to our observations in 3-week-old offspring, we detected a range of cholesteryl esters (CE), ceramides (CER), *lyso*-phosphatidylcholines (LPC), phosphatidylcholines (PC), phosphatidylethanolamine (PE) and triglycerides (TG) in adult tissues (see Supplementary Data 3 for full profiles). We observed no difference in the abundance of saturated, mono-unsaturated and poly-unsaturated lipids in the liver or adipose tissue between groups (Fig. 2h, i). However, LN offspring displayed a higher abundance of saturated lipids within

the serum when compared to the LL offspring (Fig. 2j; $P < 0.05$). Analysis of individual lipid abundance within the liver revealed 39 separate lipids to be different between the treatment groups, with an elevated abundance of diglycerides and triglycerides in LL offspring, while relative concentrations of *lyso*-phosphatidylcholines and phosphatidylcholines were elevated in LN offspring (Fig. 2k) when compared to NN offspring. Interestingly, relative lipid profiles in the gonadal fat (Fig. 2l) did not follow this pattern, with a more varied profile being observed and only 27 lipids displaying a difference between groups. Here, NL and LN tended to display a higher abundance of diglycerides and triglycerides when compared to NN and LL offspring. The most differences in lipid abundance levels were observed within the serum. Here, 57 lipids differed significantly in their relative abundance between treatment groups (Fig. 2m). LN and NL groups displayed lower levels of cholesteryl esters such as CE (20:04) and (22:06) but elevated levels of multiple phosphatidylcholines. In contrast, LL offspring displayed elevated levels of

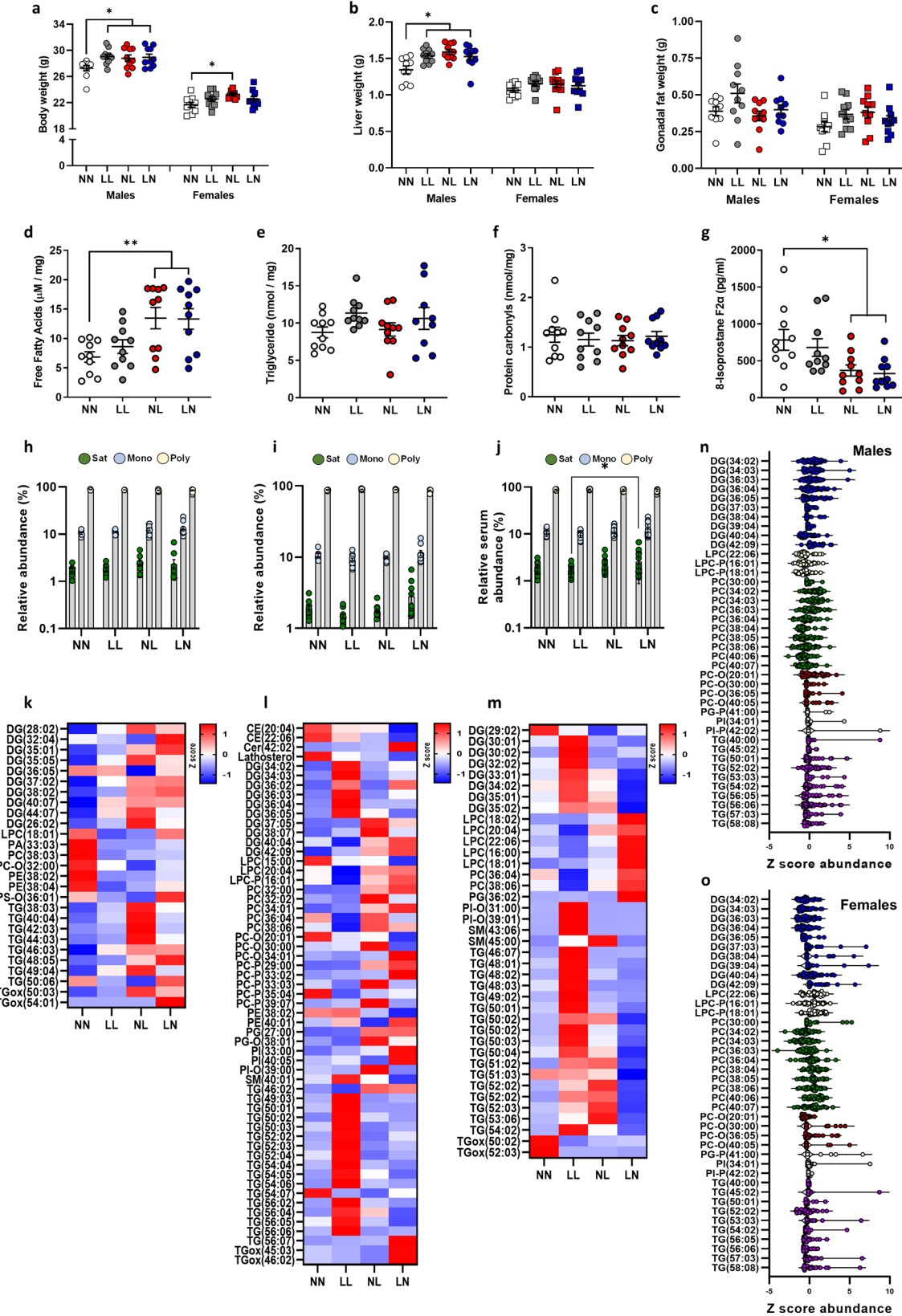

multiple diglycerides and unsaturated triglycerides when compared to the other treatment groups. These data suggest that the lipid profile of the serum may be a composite reflection of the lipids originating from different tissues. Looking at the profiles of lipid species between tissues, there appeared common patterns of elevated triglycerides in the LL offspring liver and the serum. In

contrast, NL and LN offspring displayed a comparable profile of lipids in the serum, with higher levels of diglycerides and triglycerides, matching changes seen in the gonadal fat, again with higher levels of diglycerides and triglycerides. In contrast, NL and LN offspring hepatic lipid profiles did not mirror each other, with the NL offspring matching the LL offspring more closely. As in F1

**Fig. 2 Impact of paternal diet on F1 adult-offspring weights and tissue lipid profiles.** Adult (16 weeks) male and female NN (NPD sperm and NPD seminal plasma), LL (LPD sperm and LPD seminal plasma), NL (NPD sperm and LPD seminal plasma) and LN (LPD sperm and NPD seminal plasma) offspring **a** body, **b** liver and **c** gonadal fat weight. Hepatic **d** free fatty acid and **e** triglyceride levels and serum **f** protein carbonyls and **g** isoprostane F2α levels. Relative **h** hepatic, **i** gonadal fat and **j** serum abundance of saturated (Sat) mono-unsaturated (Mono) and poly-unsaturated (Poly) lipids. Z scores of differential lipids between NN, LL, NL and LN offspring in **k** liver, **l** gonadal fat and **m** serum. Cer ceramide, DG diglyceride (water-loss product from fragmentation in source), LPC *lyso*-phosphatidylcholine, LPE lyso-phosphatidylethanolamine, LPG *lyso*-phosphatidylglycerol, PA phosphatidic acid, PC phosphatidylcholine, PC-P phosphatidylcholine plasminogen, PC-O phosphatidylcholine plasmalogen, PE phosphatidylethanolamine, PI phosphatidylinositol, SM sphingomyelin, TG triglyceride, TGox oxidised triglyceride. Differential serum lipids in **n** male and **o** female offspring. $N = 20$ offspring (10 males and 10 females) per treatment group, sampled from all litters generated. Data are expressed as mean ± SEM (**a–j**). $*p < 0.05$, $**p < 0.01$. Statistical differences were determined using a random effects regression analysis **a–g** or one-way ANOVA with Bonferoni post hoc correction **h–o**.

neonates, we observed a differential abundance of multiple lipids between males and females (Fig. 2n, o), although no overall effect of sex was observed on the global lipid profile. Here, differences were mainly restricted to specific diglyceride, phosphatidylcholines and triglyceride species, with males displaying a lower abundance of phosphatidylcholines and a higher abundance of triglyceride species.

**Impact of paternal diet on offspring liver gene expression.** As perturbed hepatic lipid metabolism underlies the development and progression of metabolic disease, we performed a transcriptomic array to define global gene expression patterns in the adult offspring (See Supplementary Data 4–6 for full gene expression profiles). Analysis of differential gene expression (>1.5 fold up/downregulation) in males and females combined revealed the smallest difference to be between the NN and LL offspring (12 genes), but the biggest difference was between the NN and NL offspring (965 genes) (Fig. 3a–d). Interestingly, LL offspring displayed a similar number of differentially expressed genes to the NN group (12 genes), while there were 567 differentially expressed genes between the NL and LN group (Fig. 3a). Separating the profiles in male and female offspring revealed similar numbers of differentially expressed genes between LL, NL, LN and NN females (Fig. 3e–h). In contrast, comparisons between males revealed only 68 differentially expressed genes between NN and LL males but 3642 genes between NN and NL males (Fig. 3i–l). Interestingly, the second biggest difference in a number of differentially expressed genes was between LL and NL males (1055 genes) (Fig. 3i).

To determine the physiological consequences of the differential gene expression, we performed gene ontology and pathway analysis on all genes that were statistically different, irrespective of their fold change (Supplementary Data 7 for full lists). Using data for males and females combined, we observed an upregulation of genes associated with 'carbohydrate/glucose metabolism', 'circulating hormone levels' and 'circulating lipid levels' in LL offspring when compared to NN offspring. Included in these were *leptin* (1.38 fold increase), *cryptochrome-1* (1.36 fold increase), *pro-glucagon* (1.28 fold increase) and *insulin receptor-related protein* (1.35 fold increased). Comparing NN with NL offspring, we observed upregulation of genes involved in cytokine signalling and neuropeptide signalling, while pathways associated with 'mRNA processing' and 'embryo development' were downregulated. Here genes such as *corticotropin-releasing factor receptor 1* (1.64 fold increase), *pro-opiomelanocortin* (1.64 fold increase), *urocortin* (1.58 fold increase), *wtap* (1.82 fold decreased), *TGF-beta receptor type-1* (1.99 fold decrease), *THO complex 2* (2.11 fold decrease) and *hypoxia inducible factor 1, alpha subunit* (1.86 fold decreased) were altered. In LN offspring, 'small molecule catabolic' and 'metabolic pathways' were upregulated. Within these pathways genes such as *acetyl-CoA carboxylase 2* (1.3 fold increase), *phosphofructokinase, liver, B-type* (1.3 fold increase) and *interleukin 1 beta* (1.59 fold

increase) were differentially expressed when compared to NN offspring. Comparing NL and LN offspring revealed pathways involved in 'metabolism', 'RNA transport' and 'protein processing in endoplasmic reticulum' to be downregulated in NL offspring. Here genes including *THO complex subunit 1* (1.65 fold downregulated) and *long-chain-fatty-acid-CoA ligase 1* (1.76 fold downregulated) were differentially expressed.

**Adult-offspring microbiome shows minimal changes in response to paternal diet.** To investigate whether adult-offspring gut bacterial diversity was affected by paternal diet (See Supplementary Data 8 for full sequencing details), we used the Faith's phylogenetic diversity and Pielou's measure of species evenness indexes. We observed no difference in bacterial diversity between groups (Fig. 4a, b). Analysis of bacterial profiles at the phylum (Fig. 4c) and family (Fig. 4d) levels showed no significant differences between treatment groups. To understand how the profile as a whole may alter, we correlated each bacterial component at the family level with each other. All groups showed a significant positive correlation between the prevalence of rikenellaceae and deferribacteraceae (Fig. 4e–g). In NN offspring, we observed a significant negative correlation between the levels of firmicutes and bacteroidetes (r = −0.854, $P = 0.007$; Fig. 4e), an association that was only present also in the NL offspring (r = −0.737, $P = 0.037$). NN offspring also showed a significant negative correlation between lachnospiraceae and S24 (r = −0.821, $P = 0.012$), and lachnospiraceae and paraprevotellaceae (r = −0.629, $P = 0.05$; Fig. 5e). Finally, NN offspring showed a positive correlation between prevotellaceae and cyanobacteria (r = 0.931, $P = 0.001$) and between Bacteroidaceae and Deferribacteres (r = 0.834, $P = 0.01$; Fig. 4e). In LL offspring, significant positive correlations were observed between deferribacteraceae and rikenellaceae, lachnospiraceae, paraprevotellaceae and bacteroidetes and between prevotellaceae, cyanobacteria and parcubacteria ($P < 0.05$; Fig. 4f). In NL offspring, significant positive correlation between deferribacteres, rikenellaceae, deferribacteraceae and bacteroidaceae were observed, while a negative association between firmicutes and bacteria was observed. Finally, in LN offspring, we observed a positive correlation between firmicutes, bacteria and lachnospiraceae, and a negative correlation between S24, lachnospiraceae and rikenellaceae.

**Changes in offspring lipid profiles are programmed across generations.** Studies have shown that offspring phenotypic programming can be propagated over multiple generations and in a paternal (male lineage)-specific manner[26,27]. Our own studies have shown that offspring cardiovascular impairments are maintained over two generations in response to parental LPD sperm and seminal plasma and transmitted down the male lineage[14]. To determine whether the differential lipid profiles observed in F1 generation offspring were also maintained within a second generation, we assayed F2 offspring liver and serum at 3 weeks of age. We observed no significant effect of offspring sex (Supplementary Fig. 3) therefore, data for F2 males and females

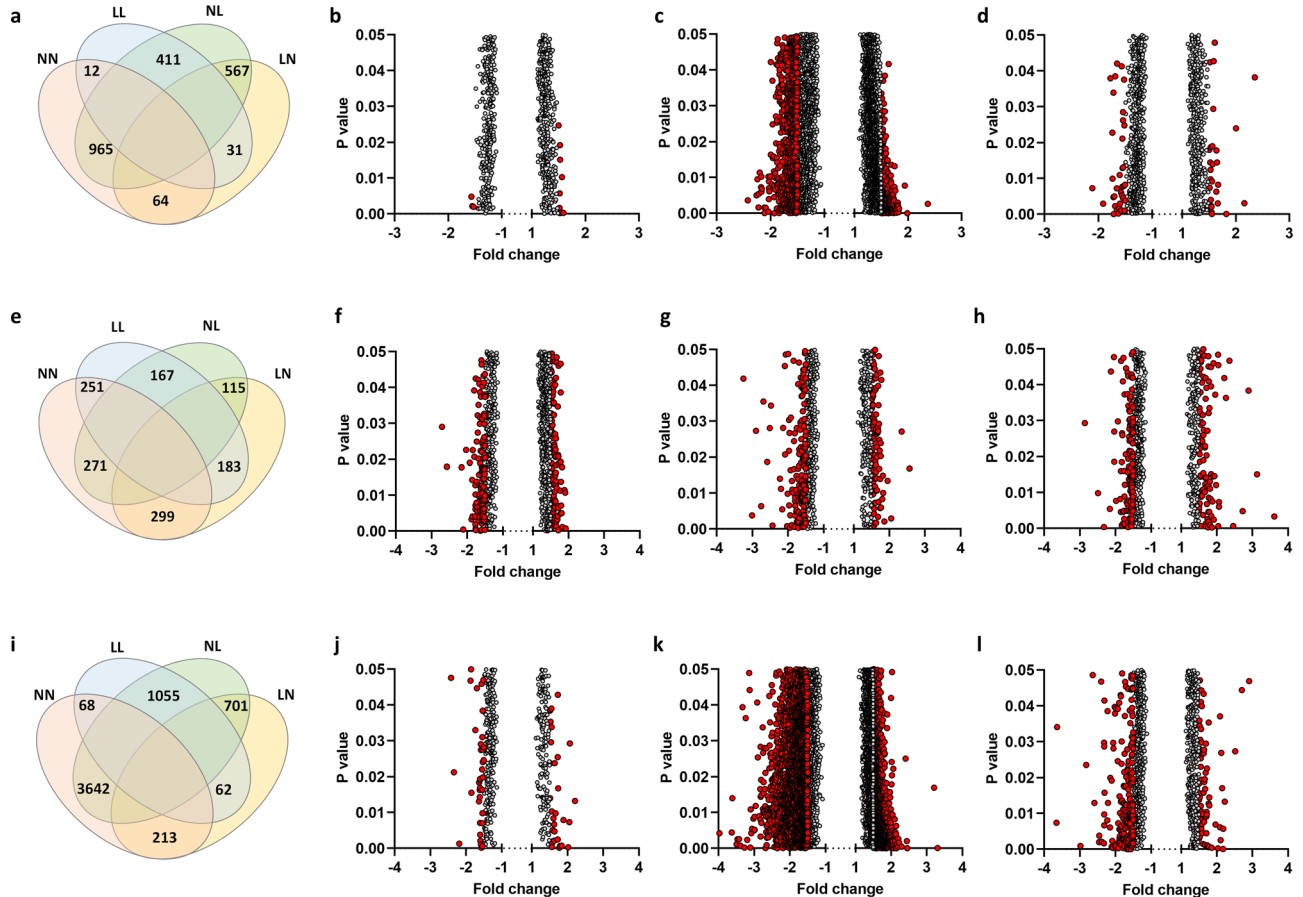

**Fig. 3 Adult-offspring hepatic gene expression profiles.** Analysis of adult (16 weeks) F1 offspring hepatic transcript profiles with **a** Venn diagram displaying number of genes with differential expression (>1.5 fold, either positive or negative) between adult (16 weeks of age) NN (NPD sperm and NPD seminal plasma), LL (LPD sperm and LPD seminal plasma), NL (NPD sperm and LPD seminal plasma) and LN (LPD sperm and NPD seminal plasma) offspring liver. Volcano plot of differential genes (>1.5 fold, either positive or negative) between **b** NN and LL adult offspring, **c** NN and NL offspring and **d** NN and LN offspring. **e** Venn diagram displaying number of genes with differential expression (>1.5 fold, either positive or negative) between adult (16 weeks of age) NN (NPD sperm and LPD seminal plasma), LL (LPD sperm and LPD seminal plasma), NL (NPD sperm and LPD seminal plasma) and LN (LPD sperm and NPD seminal plasma) female offspring liver. Volcano plot of differential genes (>1.5 fold, either positive or negative) between **f** NN and LL adult females, **g** NN and NL adult **f**emales and **h** NN and LN adult females. **i** Venn diagram displaying number of genes with differential expression (>1.5 fold, either positive or negative) between adult (16 weeks of age) NN (NPD sperm and LPD seminal plasma), LL (LPD sperm and LPD seminal plasma), NL (NPD sperm and LPD seminal plasma) and LN (LPD sperm and NPD seminal plasma) male offspring liver. Volcano plot of differential genes (>1.5 fold, either positive or negative) between **j** NN and LL adult males, **k** NN and NL adult males and **l** NN and LN adult males. $N = 4$ males and 4 females, each from separate litters. Statistical differences were determined by one-way ANOVA with Partek® Genomics Suite® analysis software.

were combined. While LL offspring displayed an elevated mean body weight when compared to NN offspring ($P = 0.01$; Fig. 5a), no other differences in body weight or liver weight were observed between groups (Fig. 5a, b).

No differences in the overall abundance of hepatic (Fig. 5c) or serum (Fig. 5d) saturated, mono-unsaturated or poly-unsaturated lipids were observed between groups. However, in contrast to F1 neonatal offspring tissues, we observed the relative abundance of 55 and 49 lipids to be significantly altered in the liver and serum respectively, between groups ($P < 0.05$; Fig. 5e, f; (see Supplementary Data 9 for full profiles). In the liver, F2 NL and LN offspring showed a relatively higher abundance of several diglycerides, sphingolipids (SM) and multiple triglycerides ($P < 0.05$; Fig. 5e), while levels of *lyso*-phosphatidylcholines (LPC) and phosphatidylcholines were relatively lower in NN and LL offspring. In contrast, in the serum, LL offspring displayed an elevated level of multiple diglycerides and phosphatidylcholines ($P < 0.05$; Fig. 5f), but comparatively lower levels of triglycerides. Apart from a few specific lipids (DG-(40:04), PG-

(36:03); PC-O(36:05), PS-(38:04)), overall levels of lipids in NL and LN offspring serum were relatively lower than in NN and LL offspring ($P < 0.05$; Fig. 5f). Interestingly, we observed few lipids to be differentially abundant between males and females ($P < 0.05$; Fig. 5g, h), which was in contrast to F1 neonates. As we observed differential lipid profiles between the ages and generations studied, with the F2 neonates more similar in differential abundance to F1 adults than F1 neonates, we compared a general hepatic lipid metric, the ratio of phosphatidylcholine to phosphatidylethanolamine (PC:PE) across all groups and ages. We observed that for all groups, the lowest ratio was seen in F1 neonatal offspring (Fig. 6). While the mean PC:PE ratio increased in NN F1 adults and F2 neonates when compared to F1 neonates, this was not significant (Fig. 6a). In LL offspring, the PC:PE ratio increased significantly in LL adults, but there was no difference between F1 and F2 neonates ($P = 0.04$; Fig. 6b). In contrast, in NL and LN offspring, the PC:PE ratio was significantly higher in both F1 adults and F2 neonatal offspring when compared to F1 neonates ($P < 0.05$; Fig. 6c, d).

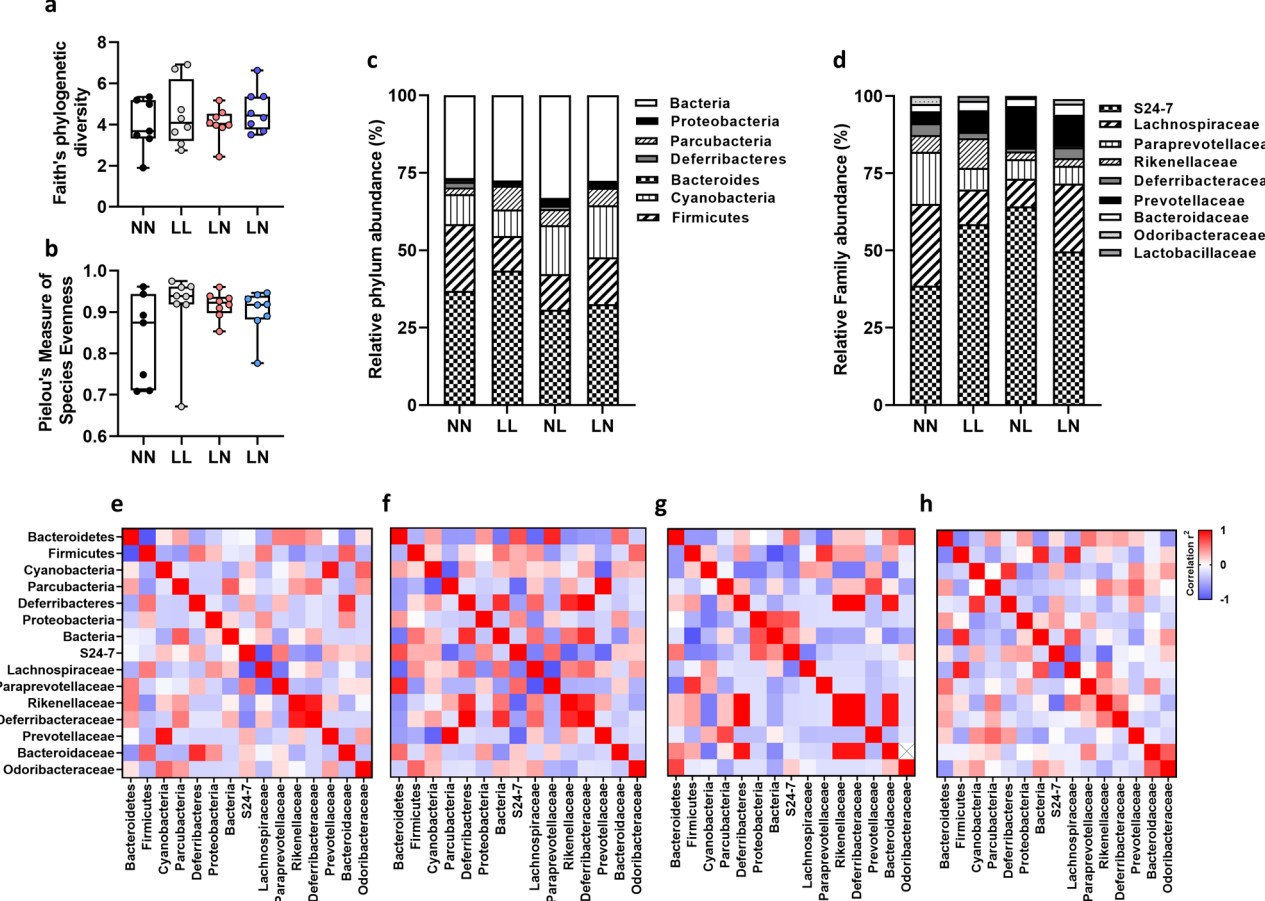

**Fig. 4 Analysis of adult-offspring gut microbiome.** Assessment of fecal bacterial profiles in using **a** Faith's phylogenetic diversity and **b** Pielou's measure of species evenness in NN (NPD sperm and NPD seminal plasma), LL (LPD sperm and LPD seminal plasma), NL (NPD sperm and LPD seminal plasma) and LN (LPD sperm and NPD seminal plasma) adult (16 weeks of age) offspring. Relative bacterial **c** phylum and **d** family proportions. Correlations between bacterial populations in **e** NN, **f** LL, **g** NL and **h** LN offspring. $N = 4$ males and 4 females, each from separate litters. Statistical differences were analysed by one-way ANOVA with Bonferoni post hoc correction **a–d** and Spearman's correlation **e–h**.

## Discussion

We have generated a mouse model in which the relative sperm and seminal plasma programming effects on offspring metabolic health across generations can be explored in response to a sub-optimal paternal LPD[13]. We observed that as offspring developed into adulthood, group-specific differences in the relative abundance of multiple lipid classes became evident. Furthermore, these changes appeared programmed in a sperm or seminal plasma-specific manner. Interestingly, unlike 3-week-old F1 offspring, who displayed comparatively few treatments group-specific differences, second-generation 3-week-old offspring possessed differential lipid abundance profiles similar in magnitude to that of the F1 adults. These data suggest that a poor paternal diet programmes offspring tissues lipid abundance over two generations, with the second-generation neonates displaying a lipid abundance phenotype reminiscent of that of an adult.

In neonatal F1 offspring, only a small number of lipids displayed a difference in abundance between treatment groups. NL offspring displayed relatively higher levels of hepatic sphingomyelin (42:00) and triglycerides (46:03; 48:04), while LN offspring showed an elevated abundance of unsaturated triglycerides. Sphingolipids have been connected to the progression of metabolic diseases due to their modulation of insulin signalling[28,29]. In addition, elevated levels of sphingolipids promote tissue inflammatory status in obese humans and mice[30,31]. Similarly, changes in the relative abundance of phosphatidylcholine and phosphatidylethanolamine have been

implicated in conditions such as NAFLD through impaired membrane integrity and endoplasmic reticulum stress[32,33]. Separately, the accumulation of hepatic diglycerides has been linked to the development of impaired insulin action[34]. While elevated hepatic diglycerides have been reported following consumption of a high sugar or high fat diet[35], all our dams and offspring were maintained on the same standard diet throughout the study, precluding a dietary origin for these differences. Alternatively, a reduction in the adipose tissue's ability to store excess lipids could account for the relative rise in liver diglycerides[36]. However, as relatively few differences in serum lipid moieties were observed, this may exclude a dysfunctional adipose tissue storage phenotype at this age. Therefore, further studies are needed to understand the origins of the changes seen in our neonatal offspring.

In contrast to neonatal offspring, analysis of adult samples revealed differential tissue- and treatment group-specific lipid profiles. Here, the serum appeared to represent a composite profile of the lipid abundance changes seen in the liver and adipose tissue. In the serum, LL offspring displayed an elevated abundance of multiple triglycerides, while NL and LN offspring displayed an elevated abundance of triglycerides and phosphatidylcholines. Elevated levels of triglycerides have been proposed as a marker of type 2 diabetes onset[37], with fatty acids of 10–18 carbons being elevated in patients with pre-diabetes and type 2 diabetes, while longer chain fatty acids (20–22 carbons) were associated with a reduced risk[37]. In contrast, type 2 diabetic

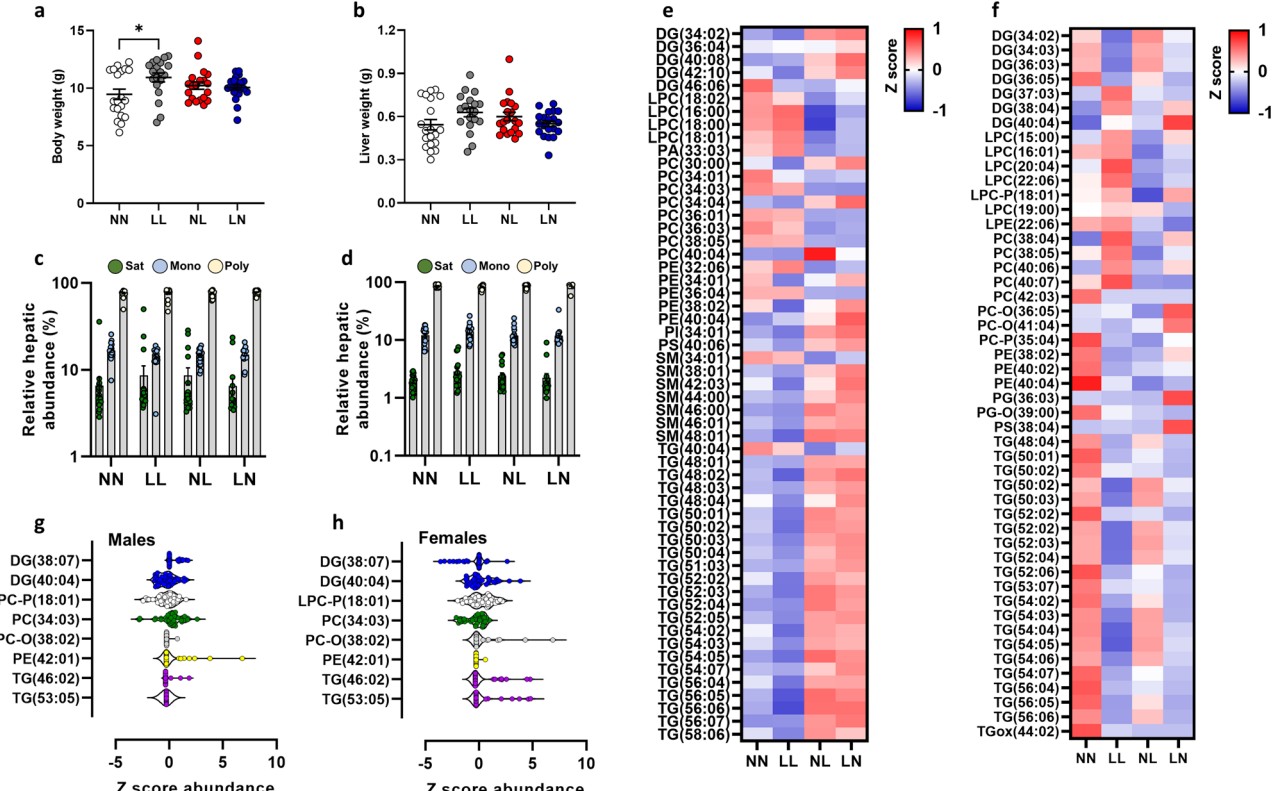

**Fig. 5 Assessment of second-generation (F2) offspring lipid profiles. a** Body weight and **b** liver weight of NN (NPD sperm and NPD seminal plasma), LL (LPD sperm and LPD seminal plasma), NL (NPD sperm and LPD seminal plasma) and LN (LPD sperm and NPD seminal plasma) F2 offspring at 3 weeks of age. Relative **c** hepatic and **d** serum abundance of saturated (Sat) mono-unsaturated (Mono) and poly-unsaturated (Poly) lipids. **e** Z scores of differential lipids between NN, LL, NL and LN offspring in **e** liver tissue and **f** serum. Cer ceramide, DG diglyceride (water-loss product from fragmentation in source), LPC lyso-phosphatidylcholine, LPE lyso-phosphatidylethanolamine, LPG lyso-phosphatidylglycerol, PA phosphatidic acid, PC phosphatidylcholine, PC-P phosphatidylcholine plasminogen, PC-O phosphatidylcholine plasmalogen, PE phosphatidylethanolamine, PI phosphatidylinositol, SM sphingomyelin, TG triglyceride, TGox oxidised triglyceride. Differential serum lipids in **g** male and **h** female offspring. N = 5–10 males and females per treatment group, selected from all litters generated. *p < 0.05. Statistical differences were determined using one-way ANOVA with Bonferoni post hoc correction.

patients showed lower levels of short-chain free fatty acids but higher levels of medium and long-chain fatty acids[37]. In LL offspring, we observed elevated abundances of triglycerides ranging from 48 (TG-(48:01)) to 68 (TG-(68:08)) carbons in length. Very-long-chain poly-unsaturated fatty acid synthesis (<24 carbons) occurs in the endoplasmic reticulum[38], with elevated levels being associated with metabolic and cardiovascular disease risk[39,40]. Additionally, LN and NL offspring displayed elevated abundances of phosphatidylcholine, including phosphatidylcholine plasmalogen (PC-O) and phosphatidylcholine plasminogen (PC-P). Phosphatidylcholine is a highly abundant lipid, contained prominently in high-density lipoprotein (HDL) particles[41] and plays a central role in hepatic metabolic homeostasis[33], regulating lipid metabolism through its activation of Peroxisome Proliferator-Activated Receptor family (PPARs)[42].

Analysis of hepatic differentially expressed genes identified comparatively few differences between NN, LL and LN offspring, while LN offspring showed a large number of differentially expressed genes when compared to all other groups. Pathways associated with carbohydrate and lipid metabolism as well as hormone levels were differentially regulated between NN and LL offspring, with *leptin, cryptochrome-1, pro-glucagon* and *insulin receptor-related protein* all increased in LL offspring. In patients with non-alcoholic fatty liver disease (NAFLD), leptin levels increase in line with the steatosis severity[43]. Separately, dysfunctional insulin secretion and insulin resistance as well as elevated serum leptin levels have been reported in patients with

NAFLD[44]. The elevated expression of *pro-glucagon* and *insulin receptor-related protein* also indicates perturbed glucose/insulin homeostasis in LL offspring. Similarly, in LN offspring, the elevated expression of *acetyl-CoA carboxylase 2* (*Acaca*) and *phosphofructokinase, liver, B-type* (*Pfkl*) is suggestive of altered regulation of central metabolic homeostasis. Acaca catalyses the carboxylation of acetyl-CoA to malonyl-CoA, performing a pivotal role in regulating the flux of carbon intermediates between carbohydrate and fatty acid metabolism. In NAFLD patients, the rate of hepatic de novo lipogenesis is increased[45], while in mice, chronic activation of acetyl-CoA carboxylase enzymes results in increased hepatic fat accumulation[46]. In NL offspring, the elevated expression of *corticotropin-releasing factor receptor 1, pro-opiomelanocortin* and *urocortin* is suggestive of metabolic stress. Corticotropin-releasing factor receptor 1 is a receptor for corticotropin-releasing factor and urocortin and elevated levels of corticotropin-releasing factor receptors and urocortin have been observed in biopsies of cirrhotic liver tissue[47]. While these observations support altered hepatic gene expression as an underlying mechanism driving the programmed dyslipidemia, further validation would be needed to confirm its role. Due to the nature of the lipidomic analyses, for many of the samples collected, the entire tissue was processed to ensure enough lipids were isolated for analysis. This limited our ability to conduct additional molecular and/or biochemical analyses on these same tissue samples. Therefore additional studies would be needed to explore the underlying mechanisms in more detail.

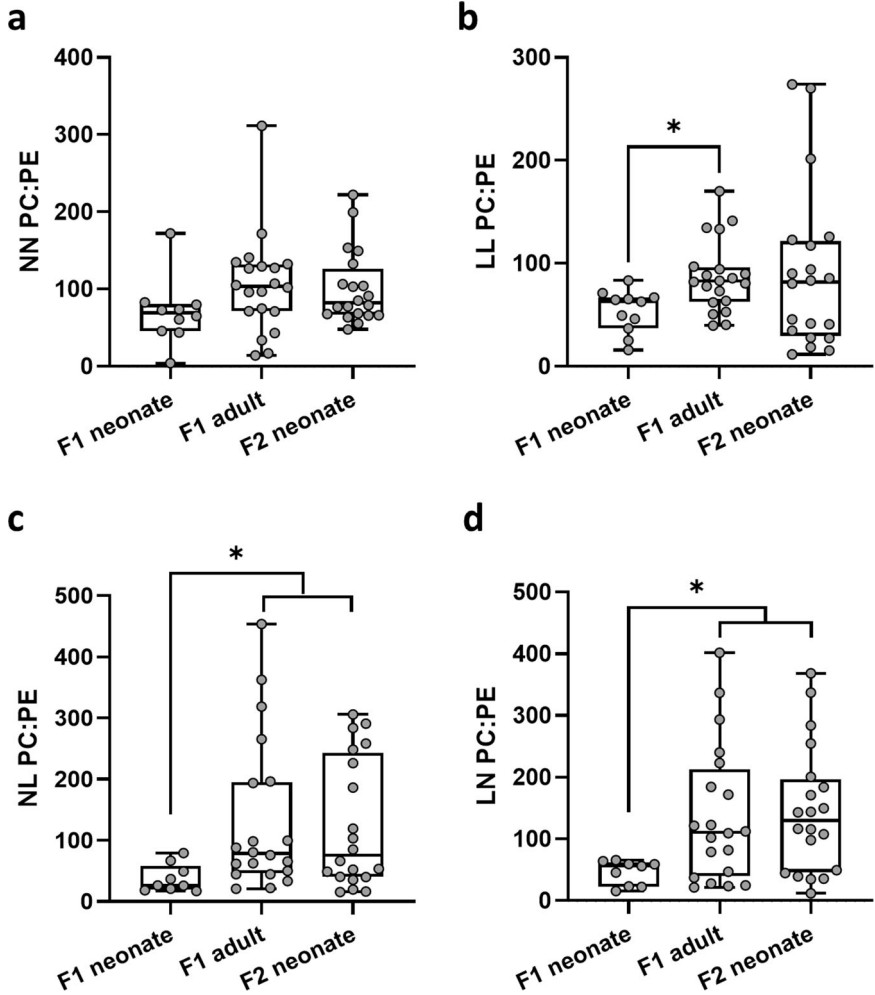

**Fig. 6 Comparison of serum lipid abundance profiles across generations.** Relative hepatic phosphatidylcholine:phosphatidylethanolamine (PC:PE) ratio in **a** NN (NPD sperm and NPD seminal plasma) F1 neonatal, F1 adult and F2 neonatal offspring, **b** LL (LPD sperm and LPD seminal plasma) F1 neonatal, F1 adult and F2 neonatal offspring, **c** NL (NPD sperm and LPD seminal plasma) F1 neonatal, F1 adult and F2 neonatal offspring and **d** LN (LPD sperm and NPD seminal plasma) F1 neonatal, F1 adult and F2 neonatal offspring. $N = 4$–10 males and females per treatment group, selected from all the litters generated. ∗$p < 0.05$. Statistical differences were determined using one-way ANOVA with Bonferoni post hoc correction.

While we observed no differences in microbial diversity or bacterial abundance between groups, we did notice that associations of bacterial abundance differed between groups. One limitation of our study is that we are unable to identify whether the changes in gut microbiome populations are a cause or a consequence of the metabolic perturbations observed. In our F1 offspring, differential tissue lipid abundance was identified from just 3 weeks of age, suggesting that any microbiome changes may occur as a downstream consequence of the paternally programmed metabolic shifts. Indeed, obesity, dyslipidaemia and glucose/intolerance, all identified in our offspring, are associated with gut dysbiosis[23]. Despite the relevance of these different bacterial combinations for offspring metabolic health being as yet undefined, disruption to the stability of the microbiota community has been associated with ill-health, including obesity[48], diabetes[49] and cardiovascular disease[50]. The significance of the microbiome in regulating lipid metabolism and health is observed through the bacterial processing of fatty acid metabolites, such as linoleic acid, by bacterial groups, including Lactobacillus. Supplementation of mice with such bacterial-derived metabolites has been shown to have beneficial effects on glucose and insulin sensitivity[51]. Furthermore, reductions in liver damage in mouse models of alcoholic liver disease have been shown following saturated long-chain fatty

acid supplementation, metabolites beneficial in supporting bacterial populations such as Lactobacillus[52]. Therefore, while we identified minimal changes in the microbiome in our offspring, further investigation is required to define the exact role of the microbiome in the paternal programming of offspring metabolic ill-health.

Previously, we have shown that a paternal low protein diet perturbed offspring cardiovascular function[14] across two generations. Changes in sperm DNA and RNA methylation, as well as sperm histone modifications and non-coding RNA have all been identified as mechanisms linking paternal well-being with offspring development[53]. Here, we observed elevated abundances of sphingolipids and triglycerides in NL and LN liver when compared to NN and LL offspring, while levels of *lyso*-phosphatidylcholines and phosphatidylcholines were lower. In contrast, levels of most of lipid groups tended to be lower in LL, NL and LN serum when compared to NN offspring. The exception here was the levels of several diglycerides, *lyso*-phosphatidylcholines and phosphatidylcholines which were elevated in LL offspring.

It is interesting to note that the differential lipid abundance profiles in second-generation neonates appeared dramatically different to those of first-generation neonates. Comparison of the

hepatic phosphatidylcholine to phosphatidylethanolamine (PC:PE) ratio revealed significant increases in the ratio in LL, NL and LN first-generation adults when compared to respective first-generation neonates. Furthermore, this ratio remained significantly higher in second-generation NL and LN offspring. In skeletal muscle, an elevated PC:PE ratio has been linked to insulin insensitivity[54], while in mitochondria, a reduction in the ratio reduces oxygen consumption and cellular ATP production, resulting in mitochondrial dysfunction and fragmentation[55]. One limitation of our second-generation analysis was that we only studied transmission down the male lineage. While paternal-specific transmission of programmed offspring ill-health has been identified in both human and animal studies[26,27], it would be of interest to define and compare the male and female-specific lineage propagation effects.

Our study gives new insight into the mechanism(s) of paternally programmed offspring metabolic ill-health. Our observations are in line with others showing that offspring from male mice fed either a low protein[56] or high fat[57] diet display perturbed metabolic phenotypes. Here, dietary-induced differential sperm non-coding RNA contents have been highlighted as the mechanism programming metabolic ill-health across generations. Our current observations also mirror our previous studies[13,14], in which offspring derived from a miss-match between the dietary background of the sperm and seminal plasma (NL and LN) displayed the largest change in metabolic phenotype, with dyslipidemia propagated into the second generation of offspring. Previously, we have hypothesised that a mismatch between the development of the preimplantation embryo (directed by the sperm) and the uterine environment (directed by the seminal plasma) has a greater impact on offspring health than when both are programmed by the semen of the same dietary origin[13]. Here, the sperm genomic/epigenomic information programs the development of the embryo prior to its arrival within a uterine environment influenced by the seminal plasma. If the programming of the sperm matches the augmentation of the uterine environment, as in the case of LL offspring, then the impact on adult health is minimised. However, in the case of NL and LN offspring, the programming of the embryo mismatches the seminal-primed uterine environment resulting in an elevated detrimental impact on offspring health. Such a hypothesis is supported by experimental data showing that a lack of seminal plasma at the time of conception is linked to adult-offspring cardio-metabolic ill-health[58]. Furthermore, a lack of maternal exposure to her partner's sperm may be of significance in the development of pregnancy complications such as preeclampsia and reduced fetal growth[59,60]. However, further studies are needed to explore these sperm and seminal programming mechanisms in detail and to understand the complex interplay between the sperm, seminal plasma, uterine environment and offspring health.

## Materials and methods

**Animal experiments**. All animal experimental procedures were conducted in accordance with UK Home Office Animal (Scientific Procedures) Act 1986 Amendment Regulations 2012, which transposed Directive 2010/63/EU into UK law and with the approval of the Animal Welfare and Ethical Review Board (AWERB) at Aston University. All animal procedures and diets have been described previously in detail[13]. Briefly, male (intact and vasectomised) 8-week-old C57BL/6 male mice (Harlan Ltd, Belton, Leicestershire, UK) were fed either a control normal protein diet (NPD; 18% casein, Special Dietary Services Ltd; UK) or isocaloric low protein diet (LPD; 9% casein, Special Dietary Services Ltd; UK) for a minimum of 8 weeks prior to use. Virgin 8-week-old female C57BL/6 mice (Charles River, UK) were superovulated (1IU pregnant mare serum gonadotrophin, 1IU human chorionic gonadotrophin, Intervet, UK) prior to artificial insemination with $10^7$ capacitated sperm isolated from NPD or LPD fed males. Sperm were isolated from the caudal epididymis into 2 ml of pre-warmed swim-out medium (135 mM NaCl, 5 mM KCl, 1 mM MgSO4, 2 mM CaCl2, 30 mM HEPES; supplemented freshly with 10 mM lactic acid, 1 mM sodium pyruvate, 20 mg/ml BSA, 25 mM NaHCO3) and left to swim up for 1 h at 37 °C. Motile sperm were collected, counted and used for non-surgical artificial insemination. Females subsequently housed overnight with a vasectomised NPD or LPD fed male. No

male (either for the provision of sperm or seminal plasma) was used more than once to generate any litter of offspring. As such, each litter was derived from separate males. Four groups of offspring were generated termed NN (NPD sperm and NPD seminal plasma), LL (LPD sperm and LPD seminal plasma), NL (NPD sperm and LPD seminal plasma) and LN (LPD sperm and NPD seminal plasma). All dams and offspring received standard chow and water ad libitum throughout the study. For the production of an F2 generation, 16-week-old F1 males ($n = 6$ males per treatment group; each from separate litters) were mated naturally to virgin, 8-week-old female C57BL/6 mice (Charles River, UK), which were acquired separately for the purpose of mating with F1 males. All dams and F2 offspring received standard chow and water ad libitum.

Male and female offspring were culled by cervical dislocation at either 3 (neonatal) or 16 (adult) weeks of age. Blood samples were centrifuged ($8000 \times g$, 4 °C, 10 min), and the serum was stored at −80 °C. Liver and gonadal adipose tissue were snap frozen and stored at −80 °C. Faecal pellets were collected from the descending colon and stored at −20 °C.

**Liver tissue assays**. Levels of liver tissue 8-isoprostane F2α (cat no. 516351 Cayman chemicals, Michigan, USA), free fatty acids (Cat no. ab65341, Abcam, Cambridge UK) and triglycerides (cat no. ab65336, Abcam, Cambridge UK) were determined by ELISA according to manufacturer's instructions. To determine liver tissue protein oxidation and nitration status, samples were homogenised in carbonate buffer and diluted to 10 μg/ml prior to analysis by ELISA (Cat no. ab210603, Abcam, Cambridge UK)[61].

**Offspring tissue lipidomic analysis**. Purified internal lipid standards were purchased from Avanti Polar lipids Inc. (Alabaster, Alabama, US). Fine chemicals and solvents were purchased from Sigma–Aldrich (Gillingham, Dorset, UK). Plastic-ware was purchased from Sarstedt (Leicestershire, UK) Chromacol (Massachusetts, United States) or Integra (Bath, Somerset, UK). Tissues were prepared, extracted and analysed in line with our previous studies[62]. Briefly, solutions of homogenised male and female, F1 and F2 liver, gonadal fat and serum, quality controls and blanks were injected into wells (96 well plate, Esslab Plate + ™, 2.4 mL/well, glass-coated) followed by a methanolic solution of internal standards (150 μL), water (500 μL) and DMT (500 μL) using a 96 channel pipette (Integra Viaflo). Internal standards consisted of deuterated representative lipids of each class. Separate quality controls were used for each tissue, made up of a pool of all samples along with homogenous stocks of human plasma, milk doped with infant formula and stock liver homogenates[63,64]. Samples were infused into an Exactive Orbitrap (Thermo, Hemel Hampstead, UK), using a TriVersa NanoMate (Advion, Ithaca US) autosampler. Samples were nanospray at 1·2 kV in the positive ion mode. The Exactive Orbitrap acquired data with a scan rate of 1 Hz (giving a mass resolution of 100,000 full widths at half-maximum at 400 $m/z$). Automatic Gain Control was set to 3,000,000 and the maximum ion injection time to 50 ms. The instrument was operated in full scan mode from $m/z$ 150–1200 Da. The lipid signals obtained are reported as relative abundance with the signal intensity of each lipid expressed relative to the total lipid signal intensity, for each sample, per mille (‰). Raw high-resolution mass-spectrometry data were processed using XCMS (www.bioconductor.org) and Peakpicker v 2.0 using an in-house R script[65] as previously described[12]. The correlation of signal intensity to the concentration of the lipid variable in QC samples (adipose tissue and liver homogenates, serum; 0.25, 0.5, 1.0×) was used to identify lipid signals that were proportional to abundance in the sample type and volume used (the threshold for acceptance was a correlation of >0.75). The quality of data collection (instrument performance) was assessed by using both QCs throughout the run and internal standards in each sample. The total signal of QCs, internal standards and samples between plates was not corrected as the average remained similarly constant throughout, with no pattern of loss or increase in signal through the data collection. Signals were then corrected (divided by the sum of signals for that sample, not including internal standards), in order to be able to compare samples in a manner unconfounded by total lipid mass. Dual spectroscopy[62] was used to interpret and finalise the lipidomic data. Specifically: [31]P NMR data of serum, adipose and liver were collected and assigned and used to determine the difference in ionisation efficiency[63].

**Adult liver transcriptomic analysis**. RNA was isolated from adult-offspring liver tissue (eight samples per treatment group; four males and females, each pair from a separate litter) using the RNeasy Plus Mini Kit (Qiagen, UK) and the TissueLyser II (Qiagen) prior to integrity assessment (Agilent 2100 Bioanalyser). Samples with an RNA integrity number (RIN) greater than 7 were used for subsequent transcript analysis. First-strand cDNA was synthesised and converted to double-stranded cDNA (GeneChip™ WT PLUS Reagent Kit (ThermoFisher Scientific, UK). cDNA purification was performed prior to fragmentation, labelling and loading onto Clariom™ S Assay Mouse GeneChip™ arrays (ThermoFisher Scientific; UK) and hybridising at 45 °C. Following washing and staining (GeneChip™ Fluidics Station 450, ThermoFisher Scientific, UK), chips were scanned (GeneChip™ Scanner 3000 7 G System, Thermo Fisher Scientific, UK).

Data were analysed with Partek® Genomics Suite® analysis software. After restricting gene lists to significant expression alterations (FDR < 0.01, $p < 0.05$), top gene lists for each dietary comparison were analysed using WEB-based

GEneSeTAnaLysis Toolkit (WebGestalt) for Gene Set Enrichment Analysis (GSEA) (http://www.webgestalt.org/) with a false detection rate (FDR) ≤ 0.05.

**Adult-offspring microbiome analysis**. DNA was isolated from F1 and F2 faecal samples using the QIAamp DNA stool mini kit (Qiagen, UK). Microbiome sequencing was conducted on the V3-V4 region of the 16 S rRNA gene following the Illumina 16 S Metagenomic Sequencing Library Preparation protocol. 16 S rRNA amplicons were generated using forward 5' (TCGTCGGCAGCGTCAGA TGTGTATAAGAGACAGCCTACGGGNGGCWGCAG) and Reverse 5' GTCT CGTGGGCTCGGAGATGTGTATAAGAGACAGGACTACHVGGGTATCT AATCC) primers, flanked by Illumina adapter-overhang sequences. Illumina dual index barcodes (Illumina XT Index Kit v2, Set A: FC-131-2001) were attached to each amplicon. AMpure XP beads (Beckman; A63882) were used for all PCR clean-up steps. Libraries were quantified using the Qubit Fluorometer and the Qubit dsDNA HS Kit (Thermo-Fisher Scientific). Library fragment-length distributions were analysed using the Agilent TapeStation 4200 and the Agilent D1000 ScreenTape Assay (Agilent; 5067–5582 and 5067–5583). Libraries were then pooled in equimolar amounts and the library pool was size-selected using the Blue Pippin (Sage Science) and a 1.5% Pippin Gel Cassette (Sage Science; BDF2010). Sequencing was performed on an Illumina MiSeq using a MiSeq Reagent Kit v3 (600 cycle) (Illumina; MS-102-3003) to generate 300 bp paired-end reads. Raw reads were processed by Qiime2 pipeline and trimmed. Greengenes version 13.8 was used in classification[66].

**Statistics and reproducibility**. Offspring body and organ weights were analysed using a multilevel random effects regression model (SPSS version 23)[13] with the paternal origin of litter incorporated as a random effect. Interactions between treatment group and offspring sex were defined, and where significant effects of offspring sex were identified, data for each sex were analysed separately and reported as such. Analysis of correlation between variables was conducted using either Spearman or Pearson correlation, depending on the normality of the data. For the analysis of tissue lipid profiles, multiple ANOVA or Kruskal–Wallis tests were used to identify lipids, or lipid classes, of significantly different abundance between groups when individual variables were normally or non-normally distributed[12]. A Benjamini–Hochberg false discovery rate analysis was applied to the analysis of individual lipids. Significance was taken at $P < 0.05$. The sample size and the number of biological replicates performed are indicated in the relevant figure legends.

**Reporting summary**. Further information on research design is available in the Nature Research Reporting Summary linked to this article.

## Data availability

All data underlying the graphs and charts presented in the main figures are provided as Supplementary Data 1 and in Excel format. All microarray data have been submitted to the Gene Expression Omnibus (GEO) at NCBI under accession number: GSE211520. All other data are available from the corresponding author on reasonable request.

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

## Acknowledgements

We would like to thank the staff at Aston University's Biomedical Research Unit for animal provision and maintenance. We would also like to thank Nazia Nazar for her assistance with depositing our microarray data. This work was supported by an Aston Research Centre for Healthy Ageing fellowship and by a Biotechnology and Biological Sciences Research Council (BBSRC) grant (BB/R003556/1) to A.J.W. and also a BBSRC grant (BB/M027252/1) awarded to A.K. to support S.F.

## Author contributions

H.L.M. prepared tissues samples for transcriptomic analysis and analysed data. S.F. conducted the lipidomics and analysed data. I.D. prepared tissues samples for hepatic metabolite analysis and analysed data. K.S. prepared tissues samples for hepatic metabolite analysis and analysed data. M.C. conducted the liver transcriptomic array and analysed data. I.K. conducted the liver transcriptomic array and analysed data. S.T.M. conducted the liver transcriptomic array and analysed data. N.H. and M.C. conducted the fecal microbiome sequencing and analysed the data. F.S. analysed the fecal microbiome sequencing data. V.W. analysed the fecal microbiome sequencing data. A.K. was involved in the methodological design for the lipidomic analysis and in the formal analysis of the data. A.J.W. was involved in the conceptualisation, data curation, formal analysis, funding acquisition, investigation, methodological design and in the writing, reviewing and editing of the manuscript. All authors reviewed and edited the manuscript

## Competing interests

The authors declare no competing interests.
