## [Peer Review File · Communications Biology]

Reviewers' comments:

Reviewer #1 (Remarks to the Author):

The publication by Morgan et al, extends on their large body of work assessing the effects of a paternal low protein diet on offspring metabolic disease risk through both sperm and seminal plasma mechanisms. In this paper they specifically assess both F1 and F2 offspring the liver tissue lipid metabolism generated from sperm and seminal plasma exposed to a low protein diet. Interestingly they found that exposure to either low protein diet sperm or seminal plasma altered hepatic metabolic and inflammatory pathways, however less of an effect was seen in the combined sperm/seminal plasma low protein group. The study is well performed, with rigorous statistical analysis and as such my comments are only of minor nature and more for my own interest, which may be therefore, be of interest to other readers.

Abstract

- Remove 'the' from components of the metabolic syndrome (line 49).

Introduction

- Remove 'will' from line 85
- More rationale as to why you assessed offspring gut microbiome and its relationship with development of metabolic disease and changes with lipid metabolism is needed.

Methods

- More details on the founders are required. For instance can you please provide N-values of the numbers of males that were used for the generation of sperm and seminal plasma? Knowing the authors work I'm assuming they use multiple founders for both sperm and seminal plasma donation.
- I think it would be beneficial to add more details on how the sperm is collected and processed. I know in previous papers you have used (135 mm NaCl, 5 mm KCl, 1 mm MgSO₄, 2 mm CaCl₂, 30 mm HEPES; supplemented freshly with 10 mm lactic acid, 1 mm sodium pyruvate, 20 mg mL⁻¹ BSA, 25 mm NaHCO₃). I'm assuming this was also the same for this publication. Do you know if injection of just the collection media modifies offspring outcomes?
- Is there a reason why you only assessed the F2 generation from the male lineage?

Results

- Give the up regulation of pathways included in circulating hormone levels including leptin. Have you considered measuring these in circulating plasma?
- Unsure what the microbiome data adds to your story or how this relates to your liver profiles. May want to consider removing.
- I know you have published the cardio-metabolic phenotype previously, but have you assessed how the lipid outcomes presented in this publication directly correlate with the same offspring metabolic phenotype, especially when you are saying that they are key drivers in many metabolic diseases. This may add validity to your study outcomes. Or at least some fasted metabolic parameters in their bloods outside of lipid profiles.

Figures

- Figure axis for the differential serum lipids are super hard to read in both figures 1 and 2 and 5. May want to consider making these larger. Same for your correlation plots in figure 4.

Discussion

- I know you briefly discuss it at the end of the discussion. But I find the phenotype of your mismatch groups which seem to be the worse effected to be so interesting. Is there any other evidence that you know of (maybe people undergoing IVF with donor sperm, but still been exposed another males seminal plasma) that has a similar experimental design and similar offspring outcomes. Because if it does then this has the potential to have huge implications for how we should treat some cases of male

factor infertility or at least inform couples of potential risks.

Reviewer #2 (Remarks to the Author):

The paper "Poor paternal diet perturbs inter-generational metabolic homeostasis in a tissue-specific manner in mice" described the effects of paternal protein restriction for a minimum of 8 weeks on the first and second generation of offspring focusing on lipid metabolism and microbiome composition. They demonstrated that the lower protein of paternal diet lead to disturbed lipid profile and gene expression and it is still remain in a second generation. The results appeared programmed in different way in sperm and seminal plasma.

The article is original and interesting, with relevant information for the field of programming of fetal development - DoHAD - regarding the paternal diet. However, some points should be highlighted.

First, the title does not fully reflect the results and analysis carried out in the article. The authors may change this to a low-protein or hypoprotein diet.

It is unclear the relationship between the offspring microbiota in the study. How would the paternal diet influence the microbiota of the offspring? Clarify this topic in the introduction section to make clearer the purpose of studying microbiota.

What is the meaning of lower abundance of higher abundance in line 202/203? It is confusing. Would it be lower abundance of PC and higher abundance of DG?

Since the results in liver gene expression are different between males and females, why not compare the differences in physiological responses between males and females?

In order to confirm the physiological changes related to lipid metabolism, it would be interesting to dose the protein expression of proteins related to lipid metabolism and inflammatory process present in chronic diseases, as well as PPARs – cytokines... confirming the changes correlated to alterations in lipid metabolism and gene expression.

In the discussion, explain the mechanism of paternal nutrition that influences the microbiota of the offspring. This analysis is not clear in the article.

The authors should add the limitations of the study.

Reviewer #3 (Remarks to the Author):

To access possible evidence that poor paternal diet can increase the risk of his offspring developing components of the metabolic syndrome, Morgan et al analyzed in the manuscript "Poor paternal diet perturbs inter-generational metabolic homeostasis in a tissue-specific manner in mice" the lipid traffic between different arrays across generations.

Differential tissue lipid abundance were observed across generations and associated with possible metabolic disorders, and the data suggested that second generation offspring displaying profiles similar to those of first generations. Additionally, these changes appeared through distinct sperm and seminal plasma-specific mechanisms.

Some minor reviews could be adjusted to better understand the study.

Results:

- 1) It would be appropriate to include in the supplementary material the analysis of quality controls that ensured the adequate performance of the analytical platforms used.
- 2) Figure 1. The lipid legends of Figures I and J are extremely small.
- 3) Line 160. It was not possible to compare the statement that "the effect on the body weight at 16 weeks of age was smaller for females than males", since in Figure 1A it was not informed whether the data are for females or males. The same is repeated for liver and weight data.
- 4) Line 165. Include in the 2D figure the data of females and males to allow the interpretation that "no effect of offspring sex was observed".
- 5) Line 221-240. It was not evident where the gene ontology and pathway analysis data are presented.
- 6) Could not find Supplementary Data
- 7) The impact of individual lipid variations between groups on metabolic pathways was very well described and compared. However, many similarities in the lipid profile were found between the 4 different groups and generations, with no separation between sex, weights, abundance of saturated, mono-unsaturated and poly-unsaturated lipids in the liver or adipose tissue, for example. Would it be possible to describe/understand why the differences in individual lipid abundance had a smaller impact on the global analysis of some factors analyzed between the 4 groups treatment between generations?

Offspring tissue lipidomic analysis

- 8) According to the Metabolomics/Lipidomics Standards Initiative, there are four levels of metabolite identification confidence.
It was not clear the level of identification of the lipids described in the manuscript.
- 9) The lipidomic analyzes were performed by direct infusion, without chromatographic separation. How were isobaric lipids accurately identified?

Reviewers' comments:

Reviewer #1 (Remarks to the Author):

The publication by Morgan et al, extends on their large body of work assessing the effects of a paternal low protein diet on offspring metabolic disease risk through both sperm and seminal plasma mechanisms. In this paper they specifically assess both F1 and F2 offspring the liver tissue lipid metabolism generated from sperm and seminal plasma exposed to a low protein diet. Interestingly they found that exposure to either low protein diet sperm or seminal plasma altered hepatic metabolic and inflammatory pathways, however less of an effect was seen in the combined sperm/seminal plasma low protein group. The study is well performed, with rigorous statistical analysis and as such my comments are only of minor nature and more for my own interest, which may be therefore, be of interest to other readers.

Response: We thank the Reviewer for assessing our manuscript and providing feedback. We have taken these comments on board and have addressed them individually below.

Abstract: Remove 'the' from components of the metabolic syndrome (line 49).

Response: We have revised the Abstract to bring it in line with the recommended word limit for Communications Biology. This has meant that reference to 'the metabolic syndrome' has now been removed.

Introduction: Remove 'will' from line 85

Response: As suggested, we have revised this sentence so that it now reads:-

'Both maternal under- and over-nutrition during pregnancy can increase the risk her adult offspring become over weight, glucose/insulin intolerant and develop dyslipidaemia [2]'.

More rationale as to why you assessed offspring gut microbiome and its relationship with development of metabolic disease and changes with lipid metabolism is needed.

Response: We agree with the Reviewers that the connection between the paternal diet, lipid profiles and the gut microbiome could be strengthened. The gut microbiota is a central regulator of host metabolism. The composition and function of the gut microbiota is dynamic and affected by physiological factors such as the amount and composition of lipids, either circulatory or from the diet. Hence, changes in central lipid abundance may influence host physiology through interaction with the gut microbiota. In our previous studies (Watkins et al. Proc Natl Acad Sci U S A. 2018 Oct 2;115 (40):10064-10069) we had used simple qPCR and identified gross changes in specific bacterial groups within LL, NL and LN offspring. These changes were in line with the general metabolic perturbations reported. In the current study, we wanted to explore these associations further.

In our current study, as we had observed significant changes in circulating and tissue lipids across our treatment groups, our intention was to determine whether changes in the gut microbiome may be occurring in parallel to the dyslipidaemia observed. Diets high in fat have been shown to alter the gut microbiome, impacting on aspects of lipid and amino acid metabolism in mice (Just et al. Microbiome.

2018;6(1):134). Separately, certain bacterial groups can metabolise lipids such as linoleic acid, producing metabolites that are beneficial in regulating glucose and insulin levels (Brown et al. J Nutr. 2003;133(10):3041–3046). Therefore, changes in gut microbiome could contribute to the dyslipidaemia seen on our mice as well as being caused by it. However, as indicated in our limitations (see response to later comment on including study limitations), we are unable to indicate whether any differences in gut bacterial abundance are contributory or a consequence of the observed dyslipidaemia.

In line with comments from this Reviewer and the other, we have strengthened the association between our lipid analyses, the metabolic disturbances and the gut microbiome in our Introduction and in the Discussion.

Methods: More details on the founders are required. For instance can you please provide N-values of the numbers of males that were used for the generation of sperm and seminal plasma? Knowing the authors work I'm assuming they use multiple founders for both sperm and seminal plasma donation.

Response: We thank the Reviewer for their comment and appreciate their request for more clarity in the details of how we generated our offspring. We can confirm that only one litter of offspring were derived from each sperm and seminal plasma donor pair. As such, no male was used to generate more than one litter in this study. If a female did not become pregnant then the same vasectomised male (seminal plasma donor) was used again. This detail has been added into the Materials and Methods (lines 568-570) which now reads:-

No male (either for the provision of sperm or seminal plasma) was used more than once to generate any litter of offspring. As such, each litter was derived from separate males.

I think it would be beneficial to add more details on how the sperm is collected and processed. I know in previous papers you have used (135 mM NaCl, 5 mM KCl, 1 mM MgSO₄, 2 mM CaCl₂, 30 mM Hepes; supplemented freshly with 10 mM lactic acid, 1 mM sodium pyruvate, 20 mg mL⁻¹ BSA, 25 mM NaHCO₃). I'm assuming this was also the same for this publication. Do you know if injection of just the collection media modifies offspring outcomes?

Response: We agree with the Reviewer and so have added in more detail in line with our previous publications. The details on the sperm collection medium have now been added into the Materials and Methods under the 'Animal experiments' section. The included text (lines 563-567) reads:-

'Sperm were isolated from the caudal epididymis into 2 ml of pre-warmed swim out medium (135 mM NaCl, 5 mM KCl, 1 mM MgSO₄, 2 mM CaCl₂, 30mM HEPES; supplemented freshly with 10 mM lactic acid, 1 mM sodium pyruvate, 20 mg/ml BSA, 25 mM NaHCO₃) and left to swim up for 1 hour at 37oC. Motile sperm were collected, counted and used for non-surgical artificial insemination'.

We thank the Reviewer for their interesting question regarding the collection medium. We have not conducted any assessment into any potential impacts of the collection medium alone. However, as all artificial inseminations were conducted using the same medium, then we would anticipate that any differences between the NN and other groups would be due to the sperm and seminal plasma composition. However, we cannot exclude the potential that the collection medium may have had some effects in its own right. As we have not studied this in specific detail, we have decided not made reference to it within the body of the manuscript.

Is there a reason why you only assessed the F2 generation from the male lineage?

Response: Our study was aimed at examining the impact of the paternal environment on the offspring. Many studies have shown parental-specific lineage transmission in offspring programming (Pembry et al Eur J Hum Genet. 2006. 14(2):159-66; Cropley et al., Mol Metab . 2016. 5(8):699-708), as such we decided to continue looking at the male lineage-specific transmission. Furthermore, these studies are in line with our other published research (Morgan et al J Physiol . 2020. 598(4):699-715) showing cardiovascular impairments are perpetuated into a second generation, again via the male lineage.

We have added additional text into the '*Changes in offspring lipid profiles are programmed across generations*' section of the Discussion to support the analysis of the male-lineage propagation. The additional text (lines 347-350) reads:-

'Studies have shown that offspring phenotypic programming can be propagated over multiple generations and in a paternal (male lineage)-specific manner [23, 24]. Our own studies have shown that offspring cardiovascular impairments are maintained over two generations in response to parental LPD sperm and seminal plasma and transmitted down the male lineage [14].'

Results: Give the up regulation of pathways included in circulating hormone levels including leptin. Have you considered measuring these in circulating plasma?

Response: We agree with the Reviewer that the analysis of such circulating factors would be of significant interest. However, the complex and untargeted lipidomic analysis conducted within our study required using a large volume of serum, meaning we did not have enough left to conduct other analyses, such as leptin levels. Furthermore, serum from sibling animals has been used in our previous publications (Watkins et al. Proc Natl Acad Sci U S A. 2018 Oct 2;115(40):10064-10069; Morgan et al., J Physiol. 2020. 598(4):699-715) meaning we do not have any remaining serum samples for validation.

Unsure what the microbiome data adds to your story or how this relates to your liver profiles. May want to consider removing.

Response: Please see our earlier comment on the association between our lipid analyses, the metabolic disturbances and the gut microbiome in our Introduction and in the Discussion. We feel that this data does have a place within this study as it offers a wider understanding of the biological processes underlying the dyslipidaemia reported in our study.

I know you have published the cardio-metabolic phenotype previously, but have you assessed how the lipid outcomes presented in this publication directly correlate with the same offspring metabolic phenotype, especially when you are saying that they are key drivers in many metabolic diseases. This may add validity to your study outcomes. Or at least some fasted metabolic parameters in their bloods outside of lipid profiles.

Response: As in response to the Reviewer's previous comments about confirming levels of circulating factors such as leptin, the complex lipidomic analysis, in combination with prior published cardiovascular, inflammatory and metabolic analyses, has precluded us from conducting any additional analyses on the serum. We have published data on fasted serum glucose levels in the F1

adults showing that the LL and NL offspring displayed impaired glucose tolerance and clearance when compared to the NN offspring (Watkins et al. Proc Natl Acad Sci U S A. 2018 Oct 2;115(40):10064-10069).

Figures: Figure axis for the differential serum lipids are super hard to read in both figures 1 and 2 and 5. May want to consider making these larger. Same for you correlation plots in figure 4.

Response: We thank the Reviewers for highlighting the difficulty in reading the text in the axes to some of our Figures. We have increased the text size to improve the clarity of them.

Discussion: I know you briefly discuss it at the end of the discussion. But I find the phenotype of your mismatch groups which seem to be the worse effected to be so interesting. Is there any other evidence that you know of (maybe people undergoing IVF with donor sperm, but still been exposed another males seminal plasma) that has a similar experimental design and similar offspring outcomes. Because if it does then this has the potential to have huge implications for how we should treat some cases of male factor infertility or at least inform couples of potential risks.

Response: We agree with the Reviewer that the greater effect size seen in the 'Mismatched' groups is of interest. At present, we are only aware of a few studies which seem to suggest some influence regarding exposure to both the sperm and seminal component may be important. The use of donor sperm in procedures such as in vitro fertilisation (IVF) has been associated with an increase in the incidence of pre-eclampsia when compared to women using their partner's sperm (Gonzalez-Comadran et al., 2014, Eur J Obstet Gynecol Reprod Biol, 182, 160-6; Wang et al., 2002. Lancet, 359, 673-4). The significance of maternal exposure to her partners sperm is supported further by observations that the risk of pre-eclampsia are increased in women who, due to impaired sperm numbers and production, had never been exposed to their partner's sperm naturally, but who received embryos fertilised using sperm surgically retrieved from their partner's testes (Wang et al., 2002). A recent large meta-analysis supports the association between the use of donor sperm and an increased risk of gestational hypertension and reduced fetal growth, however the number of studies is limited (Allen et al., Hum Reprod Update. 2021. 27(1):190-211). However, we are unaware of any human study which matches our mouse model closely, especially with respect to the long-term follow up of the offspring.

We have made more reference to such studies as these in our final section of our Discussion. We have included the following text (lines 542-546):-

'Such a hypothesis is supported by experimental data showing that a lack of seminal plasma at the time of conception is linked to adult offspring cardio-metabolic ill-health [56]. Furthermore, a lack of maternal exposure to her partner's sperm may be of significance in the development of pregnancy complications such as preeclampsia and reduced fetal growth [57, 58].'

Reviewer #2 (Remarks to the Author):

The paper "Poor paternal diet perturbs inter-generational metabolic homeostasis in a tissue-specific manner in mice" described the effects of paternal protein restriction for a minimum of 8 weeks on the first and second generation of offspring focusing on lipid metabolism and microbiome composition.

They demonstrated that the lower protein of paternal diet lead to disturbed lipid profile and gene expression and it is still remain in a second generation. The results appeared programmed in different way in sperm and seminal plasma.

The article is original and interesting, with relevant information for the field of programming of fetal development - DoHAD - regarding the paternal diet. However, some points should be highlighted.

Response: We thank the Reviewer for acknowledging the originality and interest of our study. We have addressed the Reviewer's specific comments below.

First, the title does not fully reflect the results and analysis carried out in the article. The authors may change this to a low-protein or hypoprotein diet.

Response: We appreciate the reviewer's comments on the wording of our title. While we wanted to give it a broad title, we appreciate the request for a more specific title. We have now changed this to '*Paternal low protein diet perturbs inter-generational metabolic homeostasis in a tissue-specific manner in mice*'.

It is unclear the relationship between the offspring microbiota in the study. How would the paternal diet influence the microbiota of the offspring? Clarify this topic in the introduction section to make clearer the purpose of studying microbiota.

Response: We agree with the Reviewers that the connection between the paternal diet, lipid profiles and the gut microbiome could be strengthened. The gut microbiota is a central regulator of host metabolism. The composition and function of the gut microbiota is dynamic and affected by physiological factors such as the amount and composition of lipids, either circulatory or from the diet. Hence, changes in central lipid abundance may influence host physiology through interaction with the gut microbiota. In our previous studies (Watkins et al. Proc Natl Acad Sci U S A. 2018 Oct 2;115(40):10064-10069) we had used simple qPCR and identified gross changes in specific bacterial groups within LL, NL and LN offspring. These changes were in line with the general metabolic perturbations reported. In the current study, we wanted to explore these associations further.

In our current study, as we had observed significant changes in circulating and tissue lipids across our treatment groups, our intention was to determine whether changes in the gut microbiome may be occurring in parallel to the dyslipidaemia observed. Diets high in fat have been shown to alter the gut microbiome, impacting on aspects of lipid and amino acid metabolism in mice (Just et al. Microbiome. 2018;6(1):134). Separately, certain bacterial groups can metabolise lipids such as linoleic acid, producing metabolites that are beneficial in regulating glucose and insulin levels (Brown et al. J Nutr. 2003;133(10):3041-3046). Therefore, changes in gut microbiome could contribute to the dyslipidaemia seen on our mice as well as being caused by it. However, as indicated in our limitations (see response to later comment on including study limitations), we are unable to indicate whether any differences in gut bacterial abundance are contributory or a consequence of the observed dyslipidaemia.

We have strengthened the association between our lipid analyses, the metabolic disturbances and the gut microbiome in our Introduction and in the Discussion.

What is the meaning of lower abundance of higher abundance in line 202/203? It is confusing. Would it be lower abundance of PC and higher abundance of DG?

Response: We thank the Reviewer for observing our wording error. We have revised this sentence to more correctly convey the data presented in Figure 2. This now states (lines 248-252):-

'As in F1 neonates, we observed differential abundance of multiple lipids between males and females (Fig 2N, O), although no overall effect of sex was observed on global lipid profile. Here, differences were mainly restricted to specific diglyceride, phosphatidylcholines and triglyceride species, with males displaying a lower abundance of phosphatidylcholines and a higher abundance of triglyceride species.'

Since the results in liver gene expression are different between males and females, why not compare the differences in physiological responses between males and females?

Response: We thank the Reviewer for their comment on separating the data based on sex. Central to our statistical analysis has been the inclusion of 'sex' as a fixed factor, as indicated in the 'Statistical Analysis' section of our Materials and Methods. Where offspring sex was a significant factor i.e. adult body weight, hepatic gene expression, then we have separated the data based on sex. For many of the other variables, sex was not a significant factor (as seen in the PCA plots, Supplementary Figure S1, S2 and S3) and so from a statistical perspective there was no justification in separating the data if sex was not a significant influence. As such, we have left the data analysis as currently presented.

In order to confirm the physiological changes related to lipid metabolism, it would be interesting to dose the protein expression of proteins related to lipid metabolism and inflammatory process present in chronic diseases, as well as PPARs – cytokines... confirming the changes correlated to alterations in lipid metabolism and gene expression.

Response: We agree with the Reviewer that additional follow-up studies to demonstrate the consequences of the identified lipid changes would be beneficial. Indeed, Reviewer #1 also raised this point. However, the lipidomic analysis either required the processing and use of the whole tissue sample, or rendered the tissue into a state not conducive for subsequent follow-up confirmation. Furthermore, samples from sibling animals has been used in our previous publications (Watkins et al. Proc Natl Acad Sci U S A. 2018 Oct 2;115(40):10064-10069; Morgan et al., J Physiol. 2020. 598(4):699-715) meaning we do not have sufficient remaining samples for additional validation.

In the discussion, explain the mechanism of paternal nutrition that influences the microbiota of the offspring. This analysis is not clear in the article.

Response: We have now included more detail on the justification of examining the microbiome and on the connection between offspring's metabolism and their microbiome, both in the Introduction and the Discussion. Here, we concede that we are unable to distinguish whether any change in the microbiome is a cause or a consequence of the altered lipid and central metabolic status. As such, it is possible that the paternal diet affects the microbiome indirectly through the programming of tissue dyslipaemia which affects bacterial populations.

The authors should add the limitations of the study.

Response: We agree with the Reviewer and have now included study limitations within the Discussion, where appropriate.

Reviewer #3 (Remarks to the Author):

To access possible evidence that poor paternal diet can increase the risk of his offspring developing components of the metabolic syndrome, Morgan et al analyzed in the manuscript "Poor paternal diet perturbs inter-generational metabolic homeostasis in a tissue-specific manner in mice" the lipid traffic between different arrays across generations. Differential tissue lipid abundance were observed across generations and associated with possible metabolic disorders, and the data suggested that second generation offspring displaying profiles similar to those of first generations. Additionally, these changes appeared through distinct sperm and seminal plasma-specific mechanisms.

Some minor reviews could be adjusted to better understand the study. It would be appropriate to include in the supplementary material the analysis of quality controls that ensured the adequate performance of the analytical platforms used.

Response: We agree that assessing adequate performance of the analytical platforms used is important throughout data collection. With this in mind we have clarified the use of both QCs and Internal Standards (ISs). We have added extra text into the Materials and Methods to explain our QC and Internal Standards, the additional text (lines 594-599 and 612-615) read:-

'Briefly, solutions of homogenized tissues, quality controls and blanks were injected into wells (96 well plate, Esstab Plate+™, 2·4 mL/well, glass-coated) followed by a methanolic solution of internal standards (150 µL), water (500 µL) and DMT (500 µL) using a 96 channel pipette (Integra Viaflo). Internal standards consisted of deuterated representative lipids of each class. Separate quality controls were used for each tissue, made up from a pool of all samples along with homogenous stocks of human plasma, milk doped with infant formula and stock liver homogenates...'

'The quality of data collection (instrument performance) was assessed throughout using both QCs throughout the run and internal standards in each sample. Total signal of QCs, internal standards and samples between plates was not corrected as the average remained similarly constant throughout, with no pattern of loss or increase in signal through the data collection.'

Figure 1. The lipid legends of Figures I and J are extremely small.

Response: As requested by this, and the other Reviewers, we have increased the size of the text on the X axis in Figure 1, panels I and J.

Line 160. It was not possible to compare the statement that "the effect on the body weight at 16 weeks of age was smaller for females than males", since in Figure 1A it was not informed whether the data are for females or males. The same is repeated for liver and weight data.

Response: We apologize for any confusion caused in our Figures. However, Figure 1 relates to the F1 neonates at 3 weeks of age where there was no effect of sex and so the data have been presented with the sexes combined. The data for the F1 adults is presented in Figure 2 in which the data are separated for sex, both for body weight (Fig. 2A) and liver weight (Fig. 2B). We hope this clarifies this issue.

Line 165. Include in the 2D figure the data of females and males to allow the interpretation that "no effect of offspring sex was observed".

Response: The PCA data showing that there was no significant difference between F1 neonatal males and females in the lipid abundance in either the serum or liver are contained within Supplemental Fig. 1. Please refer to this data for confirmation of no sex-effect at this age.

Line 221-240. It was not evident where the gene ontology and pathway analysis data are presented.

Response: As requested, we have now included additional Supplementary data showing the gene ontology and pathway analyses outputs from our hepatic array data. Please refer to Supplementary Data file S6.

Could not find Supplementary Data

Response: We apologize that the Reviewer could not find the attached Supplementary data. This was not an issue that was raised by either of the other Reviewers. Also, the journal did not make us aware that it had not been included in our original submission. As such, we are not sure why the Reviewer was unable to access the data. However, we assure the Reviewer that it was attached with the original submission and it will be attached with this revision.

The impact of individual lipid variations between groups on metabolic pathways was very well described and compared. However, many similarities in the lipid profile were found between the 4 different groups and generations, with no separation between sex, weights, abundance of saturated, mono-unsaturated and poly-unsaturated lipids in the liver or adipose tissue, for example. Would it be possible to describe/understand why the differences in individual lipid abundance had a smaller impact on the global analysis of some factors analyzed between the 4 groups treatment between generations?

Response: We agree that it is interesting to see the differences and similarities in lipid abundance across the generations and tissues. At this point in time we are unable to further confirm the roles of specific lipids or groups of lipids within our offspring. As in our responses to the other Reviewers, the processing of tissues for the lipidomic analysis often used up the entire sample meaning there was little, if any, left for subsequent analyses. Furthermore, many of the tissues collected from these animals have been used in previous studies publications (Watkins et al. Proc Natl Acad Sci U S A. 2018 Oct 2;115(40):10064-10069; Morgan et al., J Physiol. 2020. 598(4):699-715) again diminishing our ability to conduct further investigative analyses. As such, we are unable to confirm the role(s) of any specific lipids within our animals and which, if any, may be driving poor health. We have highlighted

in the Discussion that more work is needed to understand the origins of these lipid abundance changes and establish what their pathophysiological impact may be.

Offspring tissue lipidomic analysis: According to the Metabolomics/Lipidomics Standards Initiative, there are four levels of metabolite identification confidence. It was not clear the level of identification of the lipids described in the manuscript.

Response: We agree that there are four levels of metabolite identification confidence. The MSI level used for direct infusion mass spectrometry is level 2 (as shown in other work by us Animesh Acharjee et al., *Metabolomics*. 2017; 13(3): 25; Samuel Furse et al., *Commun Biol*. 2021; 4: 163).

The lipidomic analyzes were performed by direct infusion, without chromatographic separation. How were isobaric lipids accurately identified?

Response: We did indeed do the molecular profiling of the lipid composition using direct infusion mass spectrometry. This was done in both positive and negative ionisation modes, with an additional part of the method for fatty acid profiling. The latter was not used in this project, however collecting MS data in both positive and negative ionisation modes is useful for exactly the reason the reviewer highlights--DIMS does not include chromatographic separation of lipids and thus isobaric species will appear in the same signal. However, although species such as PC(34:1) are isobaric with PE(37:1) in positive ionisation mode, this is not the case in negative ionisation mode (in order to ionise in negative mode, PC needs an acetate or other negatively charged ion to mask the positive charge on the tetraalkylammonium group). Thus, a judicious combination of data from positive and negative ionisation modes lessens the issues around isobaric species in DIMS.

REVIEWERS' COMMENTS:

Reviewer #1 (Remarks to the Author):

I am happy with the authors changes and accept the publication in the current format.

Reviewer #2 (Remarks to the Author):

The authors have been done all revisions requested. The manuscript was improved, and it is fascinating and suitable to publish in the Communications Biology.

Reviewer #3 (Remarks to the Author):

Morgan et al analyzed in the manuscript "Paternal low protein diet perturbs inter-generational metabolic homeostasis in a tissue-specific manner in mice" the lipid traffic between different arrays across generations" some evidence that poor paternal diet can increase the risk of his offspring developing metabolic syndrome.

After the first review by the reviewers, many questions were answered, and the suggested changes were included in the manuscript.

Some minor reviews could be adjusted to complement the study and manuscript acceptance.

Line 134 and in the Materials and Methods: Complement which tissues were evaluated in the lipidomic study.

Line 482: In the sentence "Due to the nature of the lipidomic analyses, and the way in which the samples were processed" contextualize which approach the author refers to as a limitation of the study.

The study shows differential lipid abundance profiles, mainly in the second generation neonate. But it would be interesting to detail more evidence that validates how the lipid outcomes presented in this publication directly correlate with changes transferred by different generations.

Responses to Reviewers' comments:

Reviewer #1 (Remarks to the Author):

I am happy with the authors changes and accept the publication in the current format.

Response: We thank the Reviewer for taking the time to assess our manuscript and for the comments they made.

Reviewer #2 (Remarks to the Author):

The authors have been done all revisions requested. The manuscript was improved, and it is fascinating and suitable to publish in the Communications Biology

Response: We thank the Reviewer for their time and for their insightful comments on our manuscript.

Reviewer #3 (Remarks to the Author):

Morgan et al analyzed in the manuscript "Paternal low protein diet perturbs inter-generational metabolic homeostasis in a tissue-specific manner in mice" the lipid traffic between different arrays across generations" some evidence that poor paternal diet can increase the risk of his offspring developing metabolic syndrome. After the first review by the reviewers, many questions were answered, and the suggested changes were included in the manuscript.

Response: We thank the Reviewer for their time and for acknowledging that the manuscript has been revised in line the suggested changes.

Some minor reviews could be adjusted to complement the study and manuscript acceptance.

Line 134 and in the Materials and Methods: Complement which tissues were evaluated in the lipidomic study.

Response: As suggested by the Reviewer, we have indicated which tissues were used in our lipidomic analysis. Please see lines 497-498:-

'Briefly, solutions of homogenized male and female F1 and F2 liver, gonadal fat and serum, quality controls and blanks were injected into wells (96 well plate, ESSLAB Plate+™, 2·4 mL/well, glass-coated) followed by a methanolic solution of internal standards (150 µL), water (500 µL) and DMT (500 µL) using a 96 channel pipette (Integra Viaflo).'

Line 482: In the sentence "Due to the nature of the lipidomic analyses, and the way in which the samples were processed" contextualize which approach the author refers to as a limitation of the study.

Response: The main limitation here was that in order to extract a suitable quantity of lipids for analysis by LC-MS, the whole tissue sample had to be processed. This meant that there was no remaining tissue left with which we could perform additional molecular or biochemical analyses e.g. RT-qPCR. Western

Blotting, ELISA. We have included additional text at this point to highlight this limitation in more detail. Please see lines 380-384:-

'Due to the nature of the lipidomic analyses, for many of the samples collected, the entire tissue was processed to ensure enough lipids were isolated for analysis. This limited our ability to conduct additional molecular and/or biochemical analyses on these same tissue samples. Therefore additional studies would be needed to explore the underlying mechanisms in more detail.'

The study shows differential lipid abundance profiles, mainly in the second generation neonate. But it would be interesting to detail more evidence that validates how the lipid outcomes presented in this publication directly correlate with changes transferred by different generations.

Response: We agree with the Reviewer that more details on the mechanisms linking lipid metabolism across generations would be of significant interest. Studies have shown that offspring from male mice fed a low protein diet display differential expression of multiple lipid metabolism regulating genes in their livers (Carone et al., 2010. Cell. 143(7):1084-96). Furthermore, sperm non-coding RNAs isolated from the sperm of male mice fed a high fat diet are able to programme adult offspring metabolic ill-health (Chen et al., 2016. Science. 351(6271):397-400). We are currently undertaking RNA-Seq in sperm of male mice fed poor quality diets with an intention of assessing the heritability of sperm coding and non-coding RNAs to see if these are a major mechanisms. However, those studies are currently ongoing and so we are not yet in a position to publish that data.

We have added some additional text to the discussion highlighting the role sperm non-coding RNA may play in programming offspring metabolic-ill health across generations. Please see lines 430-433:-

'Our observations are in line with others showing that offspring from male mice fed either a low protein [56] or high fat [57] diet display perturbed metabolic phenotypes. Here, dietary-induced differential sperm non-coding RNA contents have been highlighted at the mechanism programing metabolic ill-health across generations'.